# Chemoselective carbene insertion into the N−H bonds of NH$_3$·H$_2$O

Zhaohong Liu[1,5], Yong Yang[1,5], Qingmin Song[1], Linxuan Li [1], Giuseppe Zanoni[2], Shaopeng Liu[1], Meng Xiang[1], Edward A. Anderson [3] & Xihe Bi [1,4] ✉

The conversion of inexpensive aqueous ammonia (NH$_3$·H$_2$O) into value-added primary amines by N−H insertion persists as a longstanding challenge in chemistry because of the tendency of Lewis basic ammonia (NH$_3$) to bind and inhibit metal catalysts. Herein, we report a chemoselective carbene N−H insertion of NH$_3$·H$_2$O using a Tp$^{Br3}$Ag-catalyzed two-phase system. Coordination by a homoscorpionate Tp$^{Br3}$ ligand renders silver compatible with NH$_3$ and H$_2$O and enables the generation of electrophilic silver carbene. Water promotes subsequent [1,2]-proton shift to generate N−H insertion products with high chemoselectivity. The result of the reaction is the coupling of an inorganic nitrogen source with either diazo compounds or N-triftosylhydrazones to produce useful primary amines. Further investigations elucidate the reaction mechanism and the origin of chemoselectivity.

Ammonia (NH$_3$) is arguably the most readily available nitrogen feedstock, with an annual production over 182 million tons from elemental nitrogen and hydrogen via the Haber−Bosch process[1,2]. The conversion of inorganic NH$_3$ or NH$_3$·H$_2$O into organic amines thus continues to attract the attention of academia and industry (Fig. 1a)[3–7]. Among those, direct catalytic syntheses of primary amines (-NH$_2$) from NH$_3$·H$_2$O attracts special attention due to the cost and step economy[8–10]. Moreover, primary amines are not only commonly found in pharmaceuticals, natural products and agrochemicals[11–15], but also could be readily derived to more complex nitrogen-containing compounds[16,17]. However, the development of such a process by transition-metal catalysis is hampered by the high strength of the N−H bond (107 kcal mol$^{-1}$) and Lewis basicity of NH$_3$, resulting in poisoning electrophilic metal catalysts[18,19]. The transition-metal-catalyzed N−H insertion reactions constitute a well-established strategy for C−N bond formation[20–29], while the few known examples of N−H insertion with NH$_3$ typically report a mixture of primary, secondary, and even tertiary amines[30,31]. Zhou and co-workers very recently disclosed a milestone progress of asymmetrical carbene insertion into the N−H bonds of NH$_3$ (in MTBE) by the cooperative action of copper complexes and chiral hydrogen-bond donor, albeit the scope of this chemistry was limited to alkyl diazoesters[32]. NH$_3$·H$_2$O is a cheaper and safer nitrogen source

than pressurized liquid NH$_3$, not requiring special equipment for transportation, storage, and handling, thus at a reduced cost per mole of NH$_3$ equivalents[10,33]. However, such a carbene insertion into the N−H bonds of NH$_3$·H$_2$O has not yet been achieved to date, presumably as the easy deactivation of transition-metal complexes by the formation of a stable Werner complex or ligand exchange with Lewis basic NH$_3$ inhibits the generation of metal-carbene complexes[18,19,32]. Moreover, even if the desired metal carbenes were generated, competitive O−H insertion with water[34,35] and multiple N−H insertion[30,31] with initially formed primary amines still exist (Fig. 1b).

Herein, we report a promising solution to this long-term challenge by Ag-catalyzed two-phase reaction system for the chemoselective carbene insertion into the N−H bonds of NH$_3$·H$_2$O using a variety of diazo compounds[24,25] and N-triftosylhydrazones[36–40] as carbene precursors (Fig. 1c). Our work stems from the following initial assumptions. First, coordination by a homoscorpionate Tp ligand protects the silver center, which enables it to react with diazo compound to generate a silver carbene even in the presence of NH$_3$[32] and water[41–43]. Second, water acts as a proton-transporter to facilitate the 1,2-proton shift of N-ylide, thereby ensuring the selectivity of carbene N−H insertion[43–46]. This process represents an efficient and practical chemoselective carbene insertion into the N−H bonds of NH$_3$·H$_2$O,

[1]Department of Chemistry, Northeast Normal University, 130024 Changchun, China. [2]Department of Chemistry, University of Pavia, Viale Taramelli 12, 27100 Pavia, Italy. [3]Chemistry Research Laboratory, University of Oxford, 12 Mansfield Road, Oxford OX1 3TA, UK. [4]State Key Laboratory of Elemento-Organic Chemistry, Nankai University, 300071 Tianjin, China. [5]These authors contributed equally: Zhaohong Liu, Yong Yang. ✉e-mail: bixh507@nenu.edu.cn

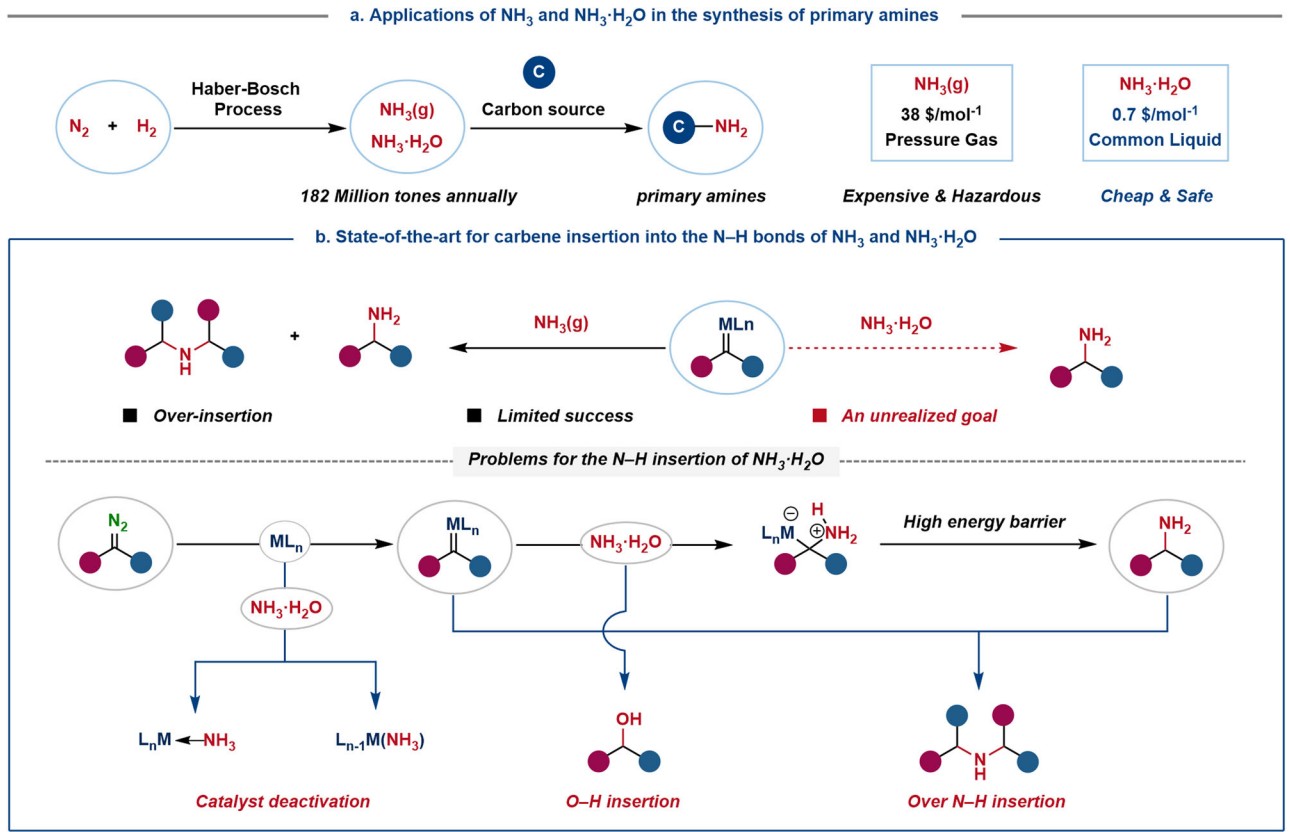

**Fig. 1 | Direct route to primary amines from inorganic NH₃ and NH₃·H₂O: strategies and challenges. a** Strategies to primary amines using $NH_3$ and $NH_3 \cdot H_2O$ as nitrogen source. **b** Challenges of N−H insertion of $NH_3 \cdot H_2O$. **c** $Tp^{Br3}Ag$-catalyzed two-phase system enables chemoselective carbene insertion into the N−H bonds of $NH_3 \cdot H_2O$. Tp, tris(pyrazolyl)borate; Ar, aryl; Tfs, 2-(trifluoromethyl) benzenesulfonyl.

enabling the efficient synthesis of primary amines, including diaryl methylamines and α-amino acid esters, valuable building blocks for pharmaceuticals and agrochemicals[11–15].

## Results and discussion
### Investigation of reaction conditions
Initial screening studies were conducted on the N−H insertion of $NH_3 \cdot H_2O$ with methyl phenyldiazoacetate **1** (Table 1). After evaluating multiple reaction parameters, the desired N−H insertion product **2** was obtained under optimized conditions in 92% yield (in 12 h from treating **1** with 8.0 equiv of $NH_3 \cdot H_2O$ at 60 °C in 1,2-dichloroethane (DCE) in the presence of 10 mol % $Tp^{Br3}Ag(thf)$)[47], along with 5% of O−H insertion product **3** (Table 1, entry 1). When $Tp^{Br3}Ag(thf)$ was replaced with $Tp^{(CF3)2}Ag(thf)$[48], the yield increased to 40% (entry 7). All other tested transition-metal catalysts failed to deliver even a trace amount of the insertion product **2**, instead leading to products of side reactions of carbene or diazo compounds (entries 8–12).

### Substrate scope
The scope of substrate diazo compounds was then explored under the optimized conditions (Fig. 2a). The tested aryl and heteroaryl diazoacetates resulted in the desired α-amino acid esters (**4–26**) in 53–98% yield, regardless of electronic character or position of the substituents on the aromatic ring. In addition to the methyl and ethyl esters, benzyl (**27**), allyl (**28**), propargyl (**29**), 2-(trimethylsilyl)ethyl (**30**), *tert*-butyl (**31**), and cyclohexyl (**32**) phenyldiazoacetates also furnished the corresponding insertion products in good yield. The reaction is not limited to donor/acceptor diazo compounds, and can be successfully expanded to donor/donor diazo compounds. A broad range of symmetric and unsymmetric diaryl diazomethanes afforded the target diaryl methylamines (**33–39**) in good to excellent yield. Unfortunately, alkyl diazo compounds are a current limitation, undergoing competing 1,2-H shift to form alkenes.

As the toxicity and potential explosivity of high-energy diazo compounds prevent scale-up of this transformation[24,25], we explored

**Table 1 | Optimization of N–H insertion of NH₃·H₂O with diazo compound**

| Entry | Cat. | Solvent | T (°C) | 2 Yield[a] | 3 Yield[b] |
|---|---|---|---|---|---|
| 1 | Tp^Br3Ag(thf) (10 mol %) | DCE | 60 | 92% | <5% |
| 2 | Tp^Br3Ag(thf) (10 mol %) | DCM | 60 | 57% | <5% |
| 3 | Tp^Br3Ag(thf) (10 mol %) | THF | 60 | N.D. | N.D. |
| 4 | Tp^Br3Ag(thf) (10 mol %) | 1,4-Dioxane | 60 | N.D. | N.D. |
| 5 | Tp^Br3Ag(thf) (10 mol %) | Toluene | 60 | 23% | N.D. |
| 6 | Tp^Br3Ag(thf) (10 mol %) | CHCl₃ | 60 | 75% | <5% |
| 7 | Tp^(CF3)2Ag(thf) (10 mol %) | CHCl₃ | 60 | 40% | N.D. |
| 8 | AgOAc (10 mol %) | CHCl₃ | 60 | N.D. | N.D. |
| 9 | Fe(TPP)Cl (10 mol %) | CHCl₃ | 60 | N.D. | N.D. |
| 10 | Rh₂(OAc)₄ (5 mol %) | CHCl₃ | 60 | N.D. | N.D. |
| 11 | Cu(OAc)₂ (10 mol %) | CHCl₃ | 60 | N.D. | N.D. |
| 12 | Pd(OAc)₂ (10 mol %) | CHCl₃ | 60 | Trace | N.D. |
| 13 | Tp^Br3Ag(thf) (10 mol %) | DCE | 80 | 75% | N.D. |

Reaction conditions: methyl phenyldiazoacetate **1** (0.3 mmol), NH₃·H₂O (2.4 mmol, 8.0 equiv) and Cat. (5–10 mol %) in solvent (4.0 mL) was stirred at 60 °C under nitrogen atmosphere for 12 h.
*N.D.* not detected, *DCE* 1,2-dichlorethane, *DCM* dichloromethane, *TPP* tetraphenylporphyrin.
[a]Isolated yield.
[b]Yield was determined by ¹H NMR with dibromomethane as the internal standard.

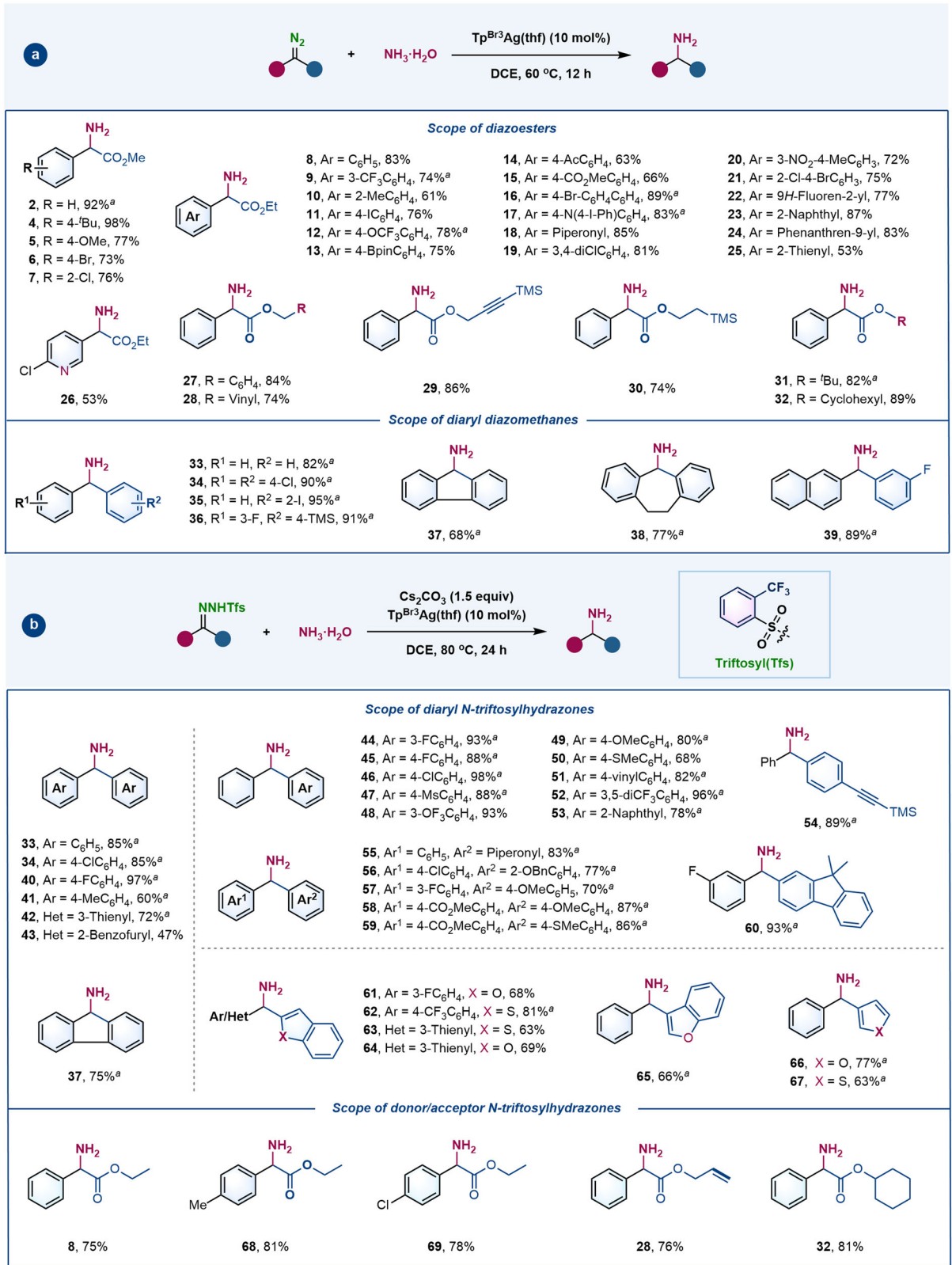

**Fig. 2 | Reaction scope of carbene insertion into the N−H bonds of NH₃·H₂O.**
**a** Silver-catalyzed N−H insertion of NH₃·H₂O with diazo compounds. **b** Silver-catalyzed N−H insertion of NH₃·H₂O with N-triftosylhydrazones. Reactions were carried out on a 0.3-mmol scale. Isolated yield reported. $^a$Isolated as hydrochloride salt. $^t$Bu tert-butyl, Bpin boronic acid pinacol ester, Ac acetyl, Ms methanesulfonyl, Ph phenyl, TMS trimethylsilyl, Bn benzyl.

**Fig. 3 | Synthetic applications. a** Gram-scale reaction. **b** Late-stage transformation of bioactive and drug molecules. [a]Started from the corresponding diazo compound. [b]Started from the corresponding *N*-triftosylhydrazone. **c** Shortened synthesis of key intermediates of drug molecule. Detailed reaction conditions are provided in the supplementary materials.

the possible use of easily prepared, bench-stable *N*-sulfonylhydrazones as carbene precursors[36,37,49–51]. Another round of optimization studies with diphenyl *N*-triftosylhydrazone as model substrate resulted in the desired diarylmethylamine (**33**) being obtained in 85% yield, when the reaction was performed in DCE at 80 °C with Cs$_2$CO$_3$ as the base (see Supplementary Table 2 for details). In contrast, *N*-tosylhydrazone proved unsuitable substrate, as the same product was obtained in a much lower 33% yield under identical conditions (entry 11, Supplementary Table 2). As shown in Fig. 2b, under these modified conditions, various diaryl *N*-triftosylhydrazones provided the desired diaryl methylamines in good to excellent yield along with a trace amount of O−H insertion products (**33**, **34**, **37**, and **40**–**60**)−the electronic and steric effects did not impact the reaction efficiency and chemoselectivity. Heteroaryl methylamines, including benzofuryl (**61**, **64**, and **65**), benzothienyl (**62** and **63**), furyl (**66**) and thienyl (**67**) methylamines, were analogously isolated in moderate to good yield from the corresponding *N*-triftosylhydrazones. Donor/acceptor *N*-triftosylhydrazones could also undergo effective N−H insertion reactions (**8**, **28**, **32**, **68**, and **69**). Notably, this in situ diazo generation protocol proved to be equally effective as the corresponding diazo-initiated reactions

(**8**, **28**, **32**–**34**, and **37**). The reaction exhibited excellent functional group tolerance of a range of functional groups, including halogen, aniline, ketone, ester, nitro, olefin, alkyne, *tert*-butyl, methoxy, trifluoromethyl and trimethylsilyl groups, predominantly providing the desired N−H insertion products along with only trace amounts of O−H insertion products.

## Gram-scale synthesis and synthetic applications
When the reaction of NH$_3$·H$_2$O with diphenyl *N*-triftosylhydrazone was conducted on a gram-scale, hydrochloride **33**·HCl was obtained pure, without chromatography, in a two-step 74% yield from diphenyl ketone (Fig. 3a). Our silver-catalyzed protocol could also be applied to late-stage modification of bioactive molecules (Fig. 3b). For instance, natural products containing a hydroxy group, such as phytol, tocopherol, vitamin D3, and (-)-β-citronellol, were first converted into the corresponding phenyldiazoacetates, then subjected to the optimized reaction conditions, affording the corresponding α-amino acid esters in moderate to good yield (**70**–**73**). Similarly, doxepin-1, a precursor of doxopin hydrochloride (a psychotropic drug) was easily converted into diarylmethylamine **74** in moderate yield through the

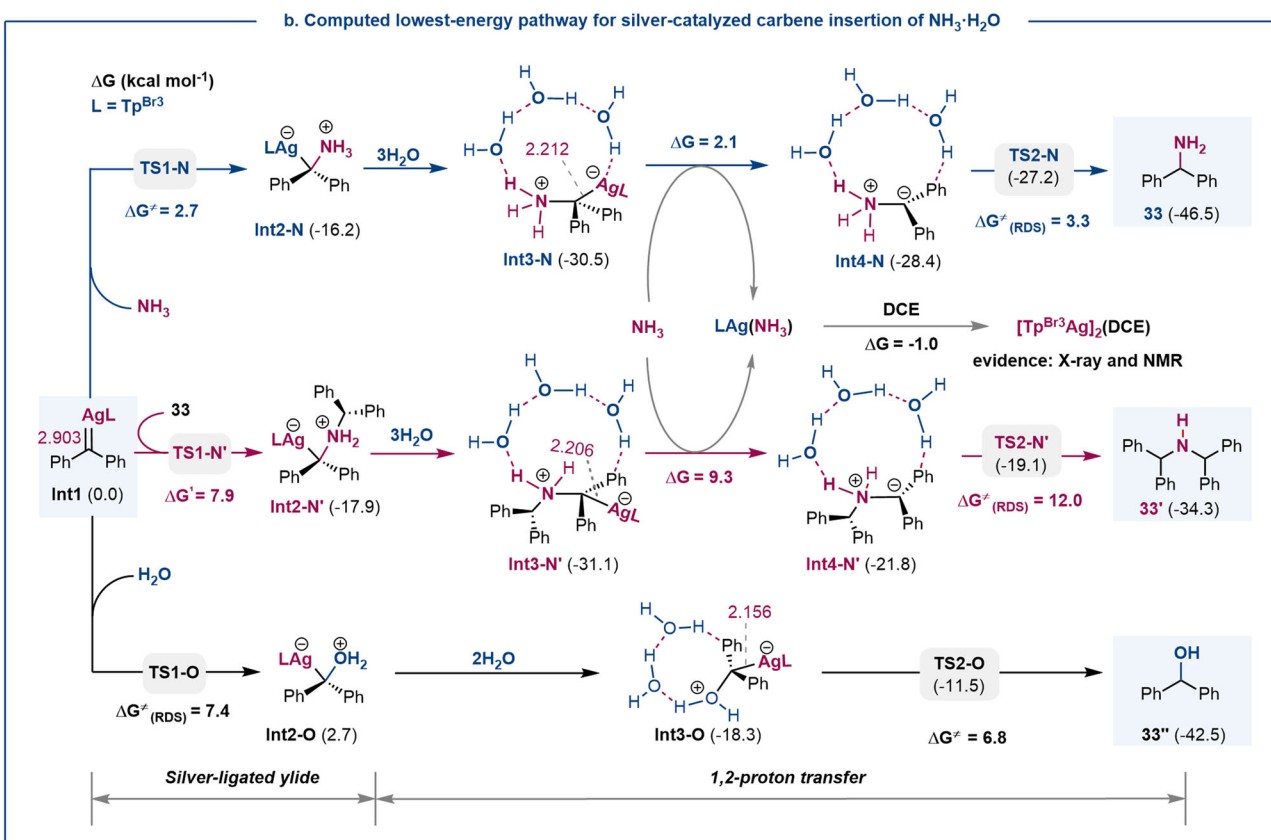

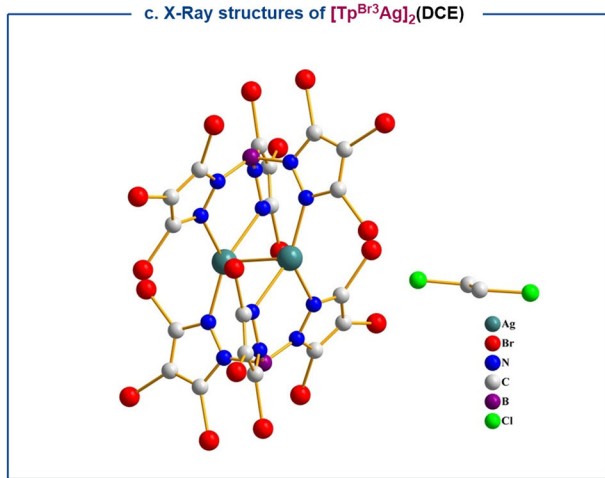

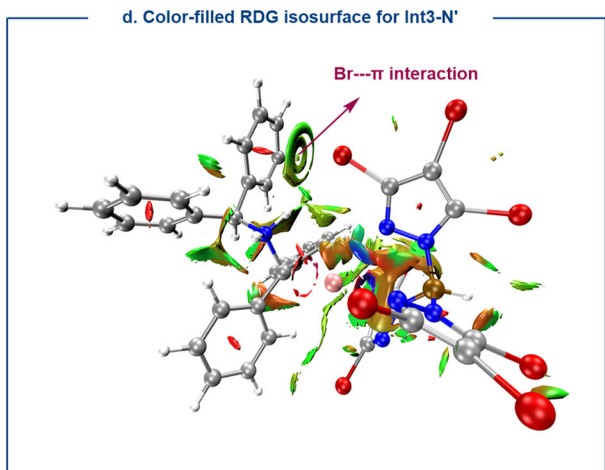

**Fig. 4 | Experimental and DFT computational insights into the mechanism.**
**a** Control experiments. **b** Computed lowest-energy pathways for silver-catalyzed carbene insertion of $NH_3$, $H_2O$, and primary amine **33**. The relative free energies present in parentheses and the (RDS) energy barriers (kcal $mol^{-1}$) were calculated at the SMD(DCE)-M06/[6-311 + G(d,p)-SDD(Ag/Br)]

level. All distances are in angstroms. **c** X-Ray structure of $[Tp^{Br3}Ag]_2$(DCE). **d** The Color-filled RDG isosurface for **Int3-N'**(isovalue set to 0.5): the water molecules are omitted for clarity: (blue) areas of attraction (covalent bonding); (green) vdW interactions; (red) areas of repulsion (steric and ring effects).

corresponding diazo compound. The diarylmethylamine moiety is present in many agrochemical and pharmaceutical compounds, for example, the hydrochloride salts of cetirizine, hydroxyzine, and meclizine possessing the structural motif of 46[12], and the antimigraine drug lomerizine containing the structural motif of 40[13]. Ketoprofen (an oral analgesic) and fenofibrate (an oral drug used to lower cholesterol levels) were converted to the corresponding primary amines 75 and 76, respectively, via the corresponding N-triftosylhydrazones. Compounds 77 and 78, intermediates in the synthesis of pharmaceutical compounds (GK-GKRP disruptor[14] and DOR1[15] agonist, respectively), could also be obtained by this N−H insertion of $NH_3 \cdot H_2O$ with N-triftosylhydrazones, showcasing the potential of our protocol for applications in drug discovery (Fig. 3c).

## Mechanistic investigations

To gain insights into the reaction mechanism and the origin of chemoselectivity, we performed control experiments and density functional theory (DFT) calculations. A competition experiment between $NH_3 \cdot H_2O$ and styrene resulted in a mixture of insertion product 33 and cyclopropane 79, suggesting that the ylide intermediate may be generated from silver carbene (Fig. 4a)[21]. DFT calculations were carried out to explore the possible reaction pathways of silver carbene Int1 with $NH_3$, diphenylmethylamine 33, and $H_2O$, respectively (see Supplementary Figs. 3–8 for details). Computed lowest-energy pathways for single N−H, double N−H, and O−H insertion are shown in Fig. 4b. The calculated Ag−C distance in Int1 of 2.093 Å points to a weak silver carbene Ag=C bond with an electrophilic carbenic carbon, which favors nucleophilic attack of X−H bonds onto Int1[21,36]. DFT calculations show that the free energy activation for the formation of silver-ligated N-ylide intermediate Int2-N from $NH_3$ (2.7 kcal mol$^{-1}$) is much lower than that for the formation of Int2-N' from primary amine 33 and Int2-O from $H_2O$ (7.9 and 7.4 kcal mol$^{-1}$, respectively). These results are consistent with the nucleophilicity of $NH_3$, primary amine 33, and $H_2O$, respectively (for the comparison of their nucleophilicity by natural population analysis, see Supplementary Fig. 9).

The weak Ag−C bond (2.212 Å) in the silver-ligated ylide favors the ligand exchange of $NH_3$ with Int3-N to release a free ylide Int4-N and silver complex LAg(NH₃). The water-assisted 1,2-proton shift from free N-ylide Int4-N opens the lowest-energy route to the N−H insertion of $NH_3$ (Supplementary Figs. 3 and 4)[43–46]. Because of the weak silver coordination, LAg(NH₃) can react with the solvent (DCE) to generate reactive [Tp$^{Br3}$Ag]₂(DCE), thus suggesting LAg(NH₃) differs from the generally stable metal−ammine complexes[8,9]. The structure of [Tp$^{Br3}$Ag]₂(DCE) was confirmed by X-ray and NMR on a sample of isolated material (Fig. 4c), which proved as effective as Tp$^{Br3}$Ag(thf) in the N−H insertion of $NH_3 \cdot H_2O$ (see Supplementary Fig. 2). These results are consistent with our initial hypothesis that the homoscorpionate Tp$^{Br3}$ ligand renders the silver catalyst compatible with $NH_3$, thus achieving the challenging N−H insertion of $NH_3 \cdot H_2O$.

Albeit similar to that of $NH_3$, the reaction pathway for the N−H insertion of diphenylmethylamine 33 (Supplementary Fig. 6) encounters a higher energy barrier for the cleavage of the Ag−C bond in ylide Int3-N' (9.3 vs. 2.1 kcal mol$^{-1}$), owing to the stronger Br···π weak interaction between the phenyl and Tp$^{Br3}$ ligand in Int3-N', as determined by color-filled reduced density gradient (RDG, Fig. 4d and Supplementary Fig. 9)[52,53]. This hypothesis was confirmed by the reaction between primary amine 33 and diphenyldiazomethane S33 without $NH_3 \cdot H_2O$, whereby in the absence of the Br···π weak interaction AgOTf led to a higher product yield (60% vs. 98%, Fig. 4a). The dual N−H insertion proceeds through an activation barrier of 12.0 kcal mol$^{-1}$, which is 8.7 kcal mol$^{-1}$ higher than that for the N−H insertion of $NH_3$, thus accounting for the absence of over N−H insertion product 33' under the optimized reaction condition. We also investigated the reaction of the silver carbene Int1 with water (Fig. S4). The Ag−C distance in Int3-O is shorter than that in Int3-N (2.156 vs. 2.212 Å), which

favors O−H insertion as the lowest-energy route for the 1,2-proton shift assisted by two molecules of water, with an activation free energy of 6.8 kcal mol$^{-1}$. Therefore, the formation of ylide Int2-O, with an activation free energy of 7.4 kcal mol$^{-1}$, is rate-limiting step for the O−H insertion.

The overall values for activation free energy for the turnover limiting-steps (1,2-H shift or O-ylide formation) of the N−H, dual N−H, and O−H insertion paths were 3.3, 12.0, and 7.4 kcal mol$^{-1}$, respectively, consistent with the experimental 10:0:1 chemoselectivity in the respective products. The weak interaction between the silver center and the carbenic carbon is beneficial to the formation and dissociation of silver-ligated ylide, which favors N−H insertion of $NH_3$ over O−H insertion. On the other hand, the significant Br···π weak interaction between the phenyl group in the substrate and the bulky Tp$^{Br3}$ ligand inhibits the dual N−H insertion.

In this work, we disclose an efficient methodology for the chemoselective carbene insertion into the N−H bonds of $NH_3 \cdot H_2O$ by a silver-catalyzed two-phase system, providing access to value-added primary amines from industrial inorganic nitrogen source. Considering the easy availability and environmental safety of $NH_3 \cdot H_2O$ and the significance of the obtained nitrogen-containing compounds, the discovery of the compatibility between the metal-carbene catalyst with the $NH_3 \cdot H_2O$ adduct may have broader applications in transition-metal-catalyzed N−H activation of $NH_3 \cdot H_2O$.

## Methods

### General procedure for silver-catalyzed N−H insertion of $NH_3 \cdot H_2O$ with diazo compounds

A Schlenk tube was charged with Tp$^{Br3}$Ag(thf) (33.0 mg, 0.03 mmol, 10 mol %). The tube was evacuated and filled with $N_2$ for three times. A mixture of $NH_3 \cdot H_2O$ (308 µL, 28–30% wt%, 0.6 mmol, 8.0 equiv) and DCE (2 mL) was injected into the tube by syringe, followed by DCE (2 mL) solution of diazo compound (0.3 mmol, 1.0 equiv). The resulting mixture was stirred at 60 °C for 12 h in the dark. When the reaction was completed, the crude reaction mixture was allowed to reach room temperature and concentrated in vacuo and purified by column chromatography on silica gel (petroleum ether/EtOAc) to afford the corresponding N−H insertion product.

### General procedure for silver-catalyzed N−H insertion of $NH_3 \cdot H_2O$ with N-triftosylhydrazones

A Schlenk tube was charged with Tp$^{Br3}$Ag(thf) (33.0 mg, 0.03 mmol, 10 mol%) and $Cs_2CO_3$ (146.6 mg, 0.45 mmol, 1.5 equiv). The tube was evacuated and filled with $N_2$ for three times. A mixture of $NH_3 \cdot H_2O$ (308 µL, 28–30% wt%, 0.6 mmol, 8.0 equiv) and DCE (2 mL) was injected into the tube by syringe, followed by DCE (2 mL) solution of N-triftosylhydrazone (0.3 mmol, 1.0 equiv). The resulting mixture was stirred at 80 °C for 24 h in the dark. When the reaction was completed, the crude reaction mixture was allowed to reach room temperature and concentrated in vacuo and purified by column chromatography on silica gel (petroleum ether/EtOAc) to afford the corresponding N−H insertion product.

## Data availability

The X-ray crystallographic coordinates for structures reported in this study have been deposited at the Cambridge Crystallographic Data Centre (CCDC), under deposition number 2166126. These data can be obtained free of charge from The Cambridge Crystallographic Data Centre via www.ccdc.cam.ac.uk/data_request/cif. The data that support the findings of this study are available within the paper and its Supplementary Information and Supplementary Data files. Raw data are available from the corresponding author on request. Materials and methods, computational studies, experimental procedures, characterization data, $^1$H, $^{13}$C, $^{19}$F NMR spectra, and mass spectrometry data are available in the Supplementary Information. Supplementary Data

File 1 contains the cartesian coordinates and energies for the computed structures.

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

## Acknowledgements

Research reported in this publication was supported by the National Natural Science Foundation of China (21871043, 1961130376), the Department of Science and Technology of Jilin Province (20200801065GH, 20190701012GH and 20180101185JC), and the Fundamental Research Funds for the Central Universities (2412022XK003, 2412019ZD001 and 2412020ZD003). X.B. and E.A. thank the Newton Trust for support (NAF\R1\191210).

## Author contributions

Z.L. and Y.Y. contributed equally to this work. Z.L., Y.Y., Q.S., L.L., S.L., and M.X. performed the experimental investigations and theoretical calculations. Z.L. and X.B. conceived the concept, designed the project, analyzed the data, and together with G.Z. and E.A. discussed the results and prepared this manuscript.

## Competing interests

The authors declare no competing interests.
