## [Peer Review File · Nature Communications]

Chemoselective carbene insertion into the N–H bonds of NH₃·H₂OREVIEWER COMMENTS

Reviewer #1 (Remarks to the Author):

In this manuscript, Bi and co-workers describe the use of a silver-based catalyst for the chemoselective insertion of a carbene group into the N-H bonds of ammonia, in a two-phase system, which leads to primary amines. The transformations have been carried out with a number of carbene precursors, ensuring a sufficient scope (Figure 2). Gram-scale preparations are also presented as well as expansion to molecules of interest for pharma. The mechanism of the reaction has been investigated by DFT calculations, showing the role of both the silver complex as well as the water solvent for the hydrogen shift step.

I find the work suitable for acceptance, with a few comments to be clarified before that:

- In the abstract: "Weak coordination by a homoscorpionate ligand enables silver compatible with NH₃ and H₂O and ensures the generation of electrophilic silver carbene". On which basis the "weak coordination" of the homoscorpionate is proposed? In any case, I do not see the need of giving this assessment.
- In the introduction, the authors should think about removing/modifying the explanation of the strategy (end of page 2) since I find some conflict in first envisioning the TpBr₃Ag catalyst but later showing catalyst screening with a number of metals.
- The authors employ "TpBr₃Ag" along the manuscript. However, this complex has been reported either as a dinuclear species [TpBr₃Ag]₂ or as a mononuclear adduct TpBr₃AgL with L as coligand. Which one is employed in this work must be indicated, and the reference of its preparation must be also added.
- In the mechanistic part, some references showing the role of water in the hydrogen-shift steps in carbene transfer reactions are missing (see for example *Organometallics* 2017, 36, 172–179; *ACS Catal.* 2014, 4, 4215–4222; *ChemCatChem* 2014, 6, 2047 – 2052)

Reviewer #2 (Remarks to the Author):

This manuscript describes a new insertion reaction of Ag carbene into N-H bond using aqueous ammonia as the nitrogen source and TpBr₃Ag as the catalyst. Though N-H insertion reaction of metal carbenoid is well-developed, using ammonia as nitrogen source is rare. Now the authors have used Ag carbene to achieve this reaction. A broad substrate scope (with >50% yields), gram-scale reactions and application on the synthesis of bioactive molecules prove the practicability of this reaction. They also carried out DFT calculations and experiments to investigate the mechanism of this reaction as well as the chemoselectivity. Overall, this manuscript is very important, and could be published after minor revision.

1. The chemical structure of the catalyst and the abbreviation of TP, TPP should be provided in the text (Table 1).
2. The authors didn't mention the alkyl ketone or alkyl diazo substrates in the text. Are these substrates compatible in this reaction?
3. In computational part, the authors have used NH₃ and H₂O solely to compare the insertion reaction, have the authors considered the real forms of NH₃·H₂O in solution?
4. Why the proton-transfer steps of N-H insertions require three H₂O molecules while O-H insertions require two H₂O molecules, could the authors provide a comparison of using one, two and three H₂O molecules?
5. The transition states of Ag-C cleavage in N-H insertions should be located, and the energy barriers of this step would be higher than 2.1 kcal/mol, which may change the rate-determining step.

REVIEWER COMMENTS

Reviewer #1 (Remarks to the Author):

In this manuscript, Bi and co-workers describe the use of a silver-based catalyst for the chemoselective insertion of a carbene group into the N–H bonds of ammonia, in a two-phase system, which leads to primary amines. The transformations have been carried out with a number of carbene precursors, ensuring a sufficient scope (Figure 2). Gram-scale preparations are also presented as well as expansion to molecules of interest for pharma. The mechanism of the reaction has been investigated by DFT calculations, showing the role of both the silver complex as well as the water solvent for the hydrogen shift step.

I find the work suitable for acceptance, with a few comments to be clarified before that:

- In the abstract: “Weak coordination by a homoscorpionate ligand enables silver compatible with NH_3 and H_2O and ensures the generation of electrophilic silver carbene”. On which basis the “weak coordination” of the homoscorpionate is proposed? In any case, I do not see the need of giving this assessment.
- In the introduction, the authors should think about removing/modifying the explanation of the strategy (end of page 2) since I find some conflict in first envisioning the $\text{Tp}^{\text{Br}^3}\text{Ag}$ catalyst but later showing catalyst screening with a number of metals.
- The authors employ “ $\text{Tp}^{\text{Br}^3}\text{Ag}$ ” along the manuscript. However, this complex has been reported either as a dinuclear species $[\text{Tp}^{\text{Br}^3}\text{Ag}]_2$ or as a mononuclear adduct $\text{Tp}^{\text{Br}^3}\text{AgL}$ with L as coligand. Which one is employed in this work must be indicated, and the reference of its preparation must be also added.
- In the mechanistic part, some references showing the role of water in the hydrogen-shift steps in carbene transfer reactions are missing (see for example *Organometallics* 2017, 36, 172–179; *ACS Catal.* 2014, 4, 4215–4222; *ChemCatChem* 2014, 6, 2047 – 2052).

Reviewer #2 (Remarks to the Author):

This manuscript describes a new insertion reaction of Ag carbene into N–H bond using aqueous ammonia as the nitrogen source and $\text{Tp}^{\text{Br}^3}\text{Ag}$ as the catalyst. Though N–H insertion reaction of metal carbenoid is well-developed, using ammonia as nitrogen source is rare. Now the authors have used Ag carbene to achieve this reaction. A broad substrate scope (with >50% yields), gram-scale reactions and application on the synthesis of bioactive molecules prove the practicability of this reaction. They also carried out DFT calculations and experiments to investigate the mechanism of this reaction as well as the chemoselectivity. Overall, this manuscript is very important, and could be published after minor revision.

1. The chemical structure of the catalyst and the abbreviation of TP, TPP should be provided in the text (Table 1).
2. The authors didn't mention the alkyl ketone or alkyl diazo substrates in the text. Are these substrates compatible in this reaction?
3. In computational part, the authors have used NH_3 and H_2O solely to comparison the insertion reaction,

have the authors considered the real forms of $\text{NH}_3 \cdot \text{H}_2\text{O}$ in solution?

4. Why the proton-transfer steps of N–H insertions require three H_2O molecules while O–H insertions require two H_2O molecules, could the authors provide a comparison of using one, two and three H_2O molecules?

5. The transition states of Ag–C cleavage in N–H insertions should be located, and the energy barriers of this step would be higher than 2.1 kcal/mol, which may change the rate-determining step.

Point-by-point response to reviewer comments

Manuscript ID: NCOMMS-22-32342

Title: Chemoselective carbene insertion into the N–H bonds of $\text{NH}_3 \cdot \text{H}_2\text{O}$

Author(s): Zhaohong Liu, Yong Yang, Qingmin Song, Linxuan Li, Giuseppe Zanoni, Shaopeng Liu, Meng Xiang, Edward A. Anderson and Xihe Bi*

Dear Reviewers,

Thank you very much for your suggestions. We have revised this manuscript according to your comments. The corrections in detail were given in the revised manuscript and were highlighted in yellow color. The detailed revision was listed as follows:

Reviewer 1:

In this manuscript, Bi and co-workers describe the use of a silver-based catalyst for the chemoselective insertion of a carbene group into the N-H bonds of ammonia, in a two-phase system, which leads to primary amines. The transformations have been carried out with a number of carbene precursors, ensuring a sufficient scope (Figure 2). Gram-scale preparations are also presented as well as expansion to molecules of interest for pharma. The mechanism of the reaction has been investigated by DFT calculations, showing the role of both the silver complex as well as the water solvent for the hydrogen shift step.

I find the work suitable for acceptance, with a few comments to be clarified before that:

Response: We really appreciate that this reviewer spent much time and great efforts on this manuscript and give affirmation to this work.

- In the abstract: “Weak coordination by a homoscorpionate ligand enables silver compatible with NH_3 and H_2O and ensures the generation of electrophilic silver carbene”. On which basis the “weak coordination” of the homoscorpionate is proposed? In any case, I do not see the need of giving this assessment.

Response: Many thanks for this valuable suggestion. Tp^{Br^3} ligand plays a critical role in protecting the activity of silver in the presence of $\text{NH}_3 \cdot \text{H}_2\text{O}$ (*Science* 2019, 366, 990–994; *Nat. Catal.* 2022, 5, 571–577), because other silver salts, rather than $\text{Tp}^{\text{Br}^3}\text{Ag}$, did not efficiently provide the desired N–H insertion products. We agree the reviewer that there is no direct and strong evidence for “weak coordination” of the homoscorpionate. Therefore, we have removed the assessment of “weak coordination” in our revised manuscript.

- In the introduction, the authors should think about removing/modifying the explanation of the strategy (end of page 2) since I find some conflict in first envisioning the $\text{Tp}^{\text{Br}^3}\text{Ag}$ catalyst but later showing catalyst screening with a number of metals.

Response: Many thanks for this valuable suggestion. According to this suggestion, we have modified the explanation of the strategy. Now it is: “First, coordination by a homoscorpionate Tp ligand protects the silver center, which enables it to react with diazo compound to generate a silver carbene even in the presence of NH_3 ³² and water.⁴¹⁻⁴³ Second, water could act as a proton-transporter to facilitate the 1,2-proton shift of N-ylide, thereby ensuring the selectivity of carbene N–H insertion.⁴³⁻⁴⁶”.

- The authors employ “Tp^{Br3}Ag” along the manuscript. However, this complex has been reported either as a dinuclear species [Tp^{Br3}Ag]₂ or as a mononuclear adduct Tp^{Br3}AgL with L as coligand. Which one is employed in this work must be indicated, and the reference of its preparation must be also added.

Response: Many thanks. Mononuclear adduct Tp^{Br3}Ag(thf) was used in our reaction, which was prepared via a known procedure (*Organometallics* **2005**, *24*, 1528-1532 and *Inorg. Chem.* **1996**, *35*, 267–268 were cited as Ref. 47 and Ref. 48). The related information has been added in the revised manuscript and Supplementary Information.

- In the mechanistic part, some references showing the role of water in the hydrogen-shift steps in carbene transfer reactions are missing (see for example *Organometallics* 2017, *36*, 172–179; *ACS Catal.* 2014, *4*, 4215–4222; *ChemCatChem* 2014, *6*, 2047 – 2052).

Response: Many thanks. The related references have been cited in the revised manuscript.

Reviewer 2 :

This manuscript describes a new insertion reaction of Ag carbene into N–H bond using aqueous ammonia as the nitrogen source and Tp^{Br3}Ag as the catalyst. Though N–H insertion reaction of metal carbenoid is well-developed, using ammonia as nitrogen source is rare. Now the authors have used Ag carbene to achieve this reaction. A board substrate scope (with >50% yields), gram-scale reactions and application on the synthesis of bioactive molecules prove the practicability of this reaction. They also carried out DFT calculations and experiments to investigate the mechanism of this reaction as well as the chemoselectivity. Overall, this manuscript is very important, and could be published after minor revision.

Response: We appreciate that this reviewer have spent much time and great efforts to review the manuscript and give affirmation to the work.

1. The chemical structure of the catalyst and the abbreviation of TP, TPP should be provided in the text (Table 1).

Response: Many thanks. According to this suggestion, the chemical structure of the catalyst and the abbreviation of TP, TPP have been provided (see Table 1 and footnote).

2. The authors didn't mention the alkyl ketone or alkyl diazo substrates in the text. Are these substrates compatible in this reaction?

Response: Many thanks for this valuable comment. When alkyl diazo compounds or *N*-trifosylhydrazones were used as substrates, only a trace amount of N–H insertion products was detected. Instead, 1,2-H shift of alkyl carbenes was observed, along with some other unidentified products. The related comments were added to the text as shown below: “Unfortunately, alkyl diazo compounds are a current limitation, undergoing competing 1,2-H shift to form alkenes.”

3. In computational part, the authors have used NH₃ and H₂O solely to comparison the insertion reaction, have the authors considered the real forms of NH₃·H₂O in solution?

Response: Many thanks for this valuable comment and suggestion. For comparison, we located the transition state (TS1-N^a) for the formation of N-ylide **Int2-N^a** from NH₃·H₂O. The activation barrier of TS1-N^a is 7.1

kcal/mol higher than that of **TS1-N** from NH_3 , thus ruling out the possibility of N–H insertion involving the real form of $\text{NH}_3\cdot\text{H}_2\text{O}$ in solution. As shown in Supplementary Fig. 2, the interaction of H_2O and NH_3 enhanced the coordination of N to Ag in **Int1-N'** (2.26 versus 3.20 Å). Therefore, the formation of N-ylide **Int2-N^a** needs to overcome a higher activation barrier (9.8 kcal/mol) to break Ag–N bond in **Int1-N^a**. In addition, only the formation of oxonium-ylide from H_2O was considered, as water is in large excess in the reaction system.

The related results and discussion have been added in revised Supplementary Information.

a. The formation of N-ylide **Int2-N** from NH_3

b. The formation of N-ylide **Int2-N^a** from $\text{NH}_3\cdot\text{H}_2\text{O}$

Supplementary Fig. 2 The comparison of the formation of N-ylides from NH_3 and $\text{NH}_3\cdot\text{H}_2\text{O}$

- Why the proton-transfer steps of N–H insertions require three H_2O molecules while O–H insertions require two H_2O molecules, could the authors provide a comparison of using one, two and three H_2O molecules?

Response: Many thanks for this valuable comment. This difference correlates with energy barriers for the proton-transfer steps of N–H and O–H insertions assisted with one, two and three H_2O molecules, respectively. As shown in Supplementary Fig. 4, the energy barriers for one, two and three H_2O molecules-assisted 1,2-proton transfer For N–H insertion are 9.1, 3.0 and 1.2 kcal/mol, respectively. By contrast, the energy barriers for H_2O -assisted 1,2-proton shift for O–H insertion are 17.9, 6.8 and 16.5 kcal/mol, respectively, as shown in Supplementary Fig. 7.

a. One H₂O molecule-assisted 1,2-proton transfer of *N*-ylide

b. Two H₂O molecule-assisted 1,2-proton transfer of *N*-ylide

c. Three H₂O molecule-assisted 1,2-proton transfer of *N*-ylide

Supplementary Fig. 4 The comparison of one, two and three H₂O molecules-assisted 1,2-proton transfer of *N*-ylide

a. One H₂O molecule-assisted 1,2-proton transfer of *O*-ylide

b. Two H₂O molecule-assisted 1,2-proton transfer of *O*-ylide

c. Three H₂O molecule-assisted 1,2-proton transfer of *O*-ylide

Supplementary Fig. 7 The comparison of one, two and three H₂O molecules-assisted 1,2-proton transfer of *O*-ylide

5. The transition states of Ag-C cleavage in N-H insertions should be located, and the energy barriers of this step would be higher than 2.1 kcal/mol, which may change the rate-determining step.

Response: Many thanks for this valuable suggestion. We have tried to locate the corresponding transition state for Ag-C cleavage in N-H insertion, but all attempts, including reaction coordinate searches and quadratic synchronous transit techniques, led directly to the formation of Ag-C cleavage products.

Finally, we would like to show our great respect to all Referees. Your critical reviews and invaluable suggestions definitely have improved the quality of this manuscript. We hope that the revised manuscript will reach the level for publication in *Nature Communications*.

REVIEWERS' COMMENTS

Reviewer #1 (Remarks to the Author):

I have previously reviewed this manuscript as reviewer#1. The revised version contains modifications in line with all my comments. I am fully satisfied with the changes and explanations and recommend acceptance as it is.

Reviewer #2 (Remarks to the Author):

This manuscript describes an important insertion reaction of Ag carbene into N-H bond using aqueous ammonia as the nitrogen source and TpBr_3Ag as the catalyst. The present manuscript is good and suitable for publication without further revisions.